# Adult Ossabaw Pigs Prefer Fermented Sorghum Tea over Isocaloric Sweetened Water

**DOI:** 10.3390/ani13203253

**Published:** 2023-10-18

**Authors:** Catherine E. Nelson, Fadi M. Aramouni, Mikayla J. Goering, Eduarda M. Bortoluzzi, Laura A. Knapp, Diana M. Herrera-Ibata, Ka Wang Li, Rabia Jermoumi, Jane A. Hooker, Joshua Sturek, James P. Byrd, Hui Wu, Valentina Trinetta, Mouhamad Alloosh, Michael Sturek, Majid Jaberi-Douraki, Lindsey E. Hulbert

**Affiliations:** 1Animal Sciences and Industry, Kansas State University, Manhattan, KS 66506, USA; 2United States Department of Agriculture-Agriculture Resource Services Center for Grain and Animal Research, Manhattan, KS 66502, USA; 3Department of Anatomy and Physiology, Kansas State University, Manhattan, KS 66506, USA; 4CorVus Biomedical, LLC, Crawfordsville, IN 47933, USA; 5Department of Statistics, Kansas State University, Manhattan, KS 66506, USA; 6School of Medicine, Indiana University, Indianapolis, IN 46202, USA; 7Department of Mathematics, Kansas State University, Manhattan, KS 66506, USA; 8Computational Comparative Medicine, Kansas State University, Manhattan, KS 66506, USA; 9FARAD Program, Kansas State University, Olathe, KS 66061, USA

**Keywords:** sorghum, fermented, tea, porcine, taste, sensory, Ossabaw, swine, pig, flavor

## Abstract

**Simple Summary:**

Fermented sorghum tea has previously been proposed to improve satiety and health. A pig model for adult weight management and fermented sorghum tea will be used for future studies; however, the literature has indicated that pigs may reject sour or bitter flavors. Previous literature studies focused on piglets and lactating sows fed ad libitum, whereas our model is focused on male and female adult pigs with restricted calorie intake. Therefore, the objective of the current work was to determine the appetitive response to fermented sorghum tea using a novel, three-way preference maze. For this study, limit-fed adult pigs not only have high affinity for fermented sorghum tea, but they also prefer the tea over the equally sweetened control solution. This is the first preference research in many years where adult pigs do not reject sour or bitter flavors if they are formulated in fermented tea. Our novel preference maze is now commercially available; therefore, our preference test methods can be replicated across laboratories, thus impacting the scientific community that uses pigs in research. Furthermore, future work using the adult pig as a model for understanding the health benefits of fermented sorghum tea will not be impeded by the avoidance behaviors of adult pigs.

**Abstract:**

Ossabaw pigs (n = 11; 5—gilts, 6—barrows; age 15.6 ± 0.62 SD months) were exposed to a three-choice preference maze to evaluate preference for fermented sorghum teas (FSTs). After conditioning, pigs were exposed, in four sessions, to choices of white FST, sumac FST, and roasted sumac-FST. Then, pigs were exposed, in three sessions, to choices of deionized H_2_O (−control; avoidance), isocaloric control (+control; deionized H_2_O and sucrose), and blended FST (3Tea) (equal portions: white, sumac, and roasted sumac). When tea type was evaluated, no clear preference behaviors for tea type were observed (*p* > 0.10). When the 3Tea and controls were evaluated, pigs consumed minimal control (*p* < 0.01;18.0 ± 2.21% SEM), and they consumed great but similar volumes of +control and 3Tea (96.6 and 99.0 ± 2.21% SEM, respectively). Likewise, head-in-bowl duration was the least for −control, but 3Tea was the greatest (*p* < 0.01; 5.6 and 31.9 ± 1.87% SEM, respectively). Head-in-bowl duration for +control was less than 3Tea (*p* < 0.01; 27.6 vs. 31.9 ± 1.87% SEM). Exploration duration was the greatest in the area with the −control (*p* < 0.01; 7.1 ± 1.45% SEM), but 3Tea and +control exploration were not different from each other (1.4 and 3.0 ± 1.45% SEM, respectively). Regardless of tea type, adult pigs show preference for FST, even over +control. Adult pigs likely prefer the complexity of flavors, rather than the sweetness alone.

## 1. Introduction

Sorghum is a potentially more sustainable crop than other grains due to the plant’s resilience to dry, hot climates [1,2,3]. Therefore, new, palatable products need to be developed to improve both food availability and nutrition security. Compared to other cereal grains, sorghum may provide more nutrition security because of greater concentrations of phenolic compounds [4,5]. Phenolic compounds possess significant antioxidant abilities that modulate chronic inflammation [6], prevent cancerous cells proliferation [7], and may even improve satiety during weight loss programs involving tea-based products [8].

Nonetheless, sorghum strains with high concentrations of phenolic compounds (e.g., red or black sorghum) may contribute to increased bitterness in products because of the associated increased tannins [5,9,10]. Furthermore, when sorghum is heated, it decreases protein bioavailability in cooked sorghum, which may be caused by phenolic compounds binding to proteins [5,11]. These drawbacks may be remedied through the process of fermentation. [12,13,14]. For example, sorghum can be cultured to produce potentially beneficial byproducts and increase protein bioavailability, and the fermentation of the liquid may be palatable to adults [15,16,17].

For this current project, fermented sorghum-based tea was created because of the potential to have high concentrations of antioxidants and probiotics, thus providing a nutrition-secure beverage. Although sorghum teas and fermented teas originated in Northeast China around 220 BC [18,19], fermented teas are new to other countries. Kombucha is a recent example of a commercialized fermented tea in the U.S., which is produced by first seeping black, green, or oolong tea leaves, and then adding sugar and a symbiotic culture of bacteria and yeast (SCOBY) [18,20]. These same methods could be applied to sorghum, but when a kombucha formula is applied instead, the fermentation results in sour flavors and an astringent mouthfeel [17,20]. A major influence of kombucha flavor is the organic acid content produced by the microorganisms within the SCOBY. Specifically, malic, gallic, and lactic acids contribute to added sour flavors and a dry mouth feel, which are also associated with red wines even though these compounds contribute to the antioxidant properties of fermented beverages [20,21,22].

Although the final product will be primarily marketed for humans, the current authors anticipate that the byproducts of producing fermented sorghum may also improve health among commercially produced pigs, particularly among adult pigs that are fed restricted calories to avoid obesity. However, previous research shows that pigs may reject sour or bitter flavors. Nevertheless, most preference research has been conducted on either young commercial pigs at weaning, or ad libitum-fed sows that have inappetence after parturition [23,24,25,26]. There is a lack of research publications on flavor preference in adult pigs that must be limit fed.

Therefore, the objectives of this study were: (1) to assess adult pig preference among three types of formulated sorghum teas, and (2) to compare the preference or avoidance behaviors among choices of blended fermented sorghum tea, positive control isocaloric sugar water, and negative control water. The hypothesis was that adult pigs would prefer the sorghum tea with the least amount of gallic acid and the isocaloric control (sugar water) over a blend of fermented sorghum teas.

## 2. Materials and Methods

### 2.1. Animals and Housing

This study followed the guidelines set forth by the Guide for the Care and Use of Agricultural Animals in Agriculture and Teaching (CorVus IACUC #4745). Sample size calculation was completed using variables from a preference study in another line of Iberian minipigs [27]. From this research, volume intake, duration of head-over-bowl, exploration, and consumption were the main variables considered. Then, sample size calculations were made using veterinary epidemiology methods previously described [28]. The calculator was set to an alpha of 0.05 and the power (1 beta) to 95. Each measure’s standard error was entered and the desired magnitude of difference between genders for each measure was entered. The average sample estimation was 5 pigs per gender. Behavioral tests and pig types from the literature were not an exact match for this experiment; therefore, an additional pig per gender was requested, resulting in 12 subjects. After IACUC approval, pigs were carefully selected for our project. Future studies aim to better understand the health benefits and behavioral changes associated with the consumption of sorghum tea, especially among adult subjects. The Corvus research herd had up to 200 pigs, mainly breeding dams, sires, and offspring. Criteria for this project were healthy, lean adult pigs that had never reproduced and were over 8 months old. Therefore, we were limited to virgin adult female pigs and barrows that already had familiarity with each other for ease of acclimation into new pairs and the experiment facility. The animal portion of the project was conducted between May and June 2022. Twelve adult Ossabaw gilts and barrows (age 15.6 ± 0.62 SD months; 53.5 ± 0.62 kg body weight) were pair-housed by gender at indoor, climate-controlled facilities (21 °C; CorVus Biomedical, LLC, Crawfordsville, Indiana). Artificial lighting was a 12 h cycle, with lights on from 0800 to 2000 h. One month prior to the start of the experiment, pig pairs were randomized into one of six pens that were stratified by gender (gilts and barrows). Pens provided 14.92 m^2^ of free space with plastic slotted flooring (Maxima Sow Slat Floor, Greenfield, WI, USA). The gating for the pens were 94.0 cm tall and custom made with vertical standard stainless bars (Thorp Equipment, Thorp, WI, USA). Two feeders (S3 Series Dry Sow Feeder, Crystal Spring Hog Equipment Agathe, Manitoba, CA) were provided on each side of the pen with a drinker (ad libitum well water; QC Swine Water Cup System, Delphi, IN, USA) next to one of the feeders. Environmental enrichment consisted of two combined stainless steel chains, with one link fastened near the drinker. Pigs were fed according to National Research Council (NRC) standards. Pigs were fed a total of 750 g feed per day to maintain a healthy, lean body type (15% Crude Protein, 3% Crude Fat; 5% Fiber; DuMOR Hog Grower, Tractor Supply, Brentwood, TN, USA). Pigs were fed at 0900 h and each daily testing sessions took place one hour later. Pig pairs were randomly assigned to pen using the RAND function of Excel (Microsoft Excel, Version 10; Redmond, WA, USA). Then, individual pigs within each pen were randomized for order of entry into the testing arena.

### 2.2. Formulation of Solutions

Three varieties of sorghum (Nu Life, Scott City, KS, USA) were used for this study: raw white, raw sumac, and roasted sumac strains. To create the sorghum tea (Figure 1), commercially available (for human consumption) sorghum was steeped in sterile water at room temperature for 16 h. Then, the supernatant was recovered by filtering and brought to a boil before sugar was dissolved (Figure 1). Once boiled, 2 cups of supernatant (473 mL) were added to a jar with 1 cup of sugar (200 g) and 8 cups (1892 mL) of deionized water. Afterward, the solution was cooled to room temperature, and inoculated with a commercially available culture (Symbiotic Culture of Bacteria and Yeast, SCOBY; The Kombucha Shop, Madison, WI, USA). Then, the solution fermented for 7 days at room temperature. Fermentation was then stalled by refrigeration. For this current experiment, the solution was refrigerated for 72 h before use. The pH of the fermented solutions were monitored and averaged 3.12 ± 0.12 SD. An isocaloric (0.288 calories per mL), positive control (+control) was created with sugar (sucrose) and boiled distilled water. The negative control (−control) was distilled water only. Each batch of solution was sampled and frozen at −20 °C for gallic acid analyses. Gallic analyses were determined using the Folin–Ciocalteu method as described previously [29,30,31,32].

### 2.3. Preference Testing

A commercially available, 3-compartment (i.e., pod) preference maze was used for the preference test (3-PodShape, by Hulbert and ShapeMaster, Odgen, IL, USA; Figure 2; Appendix A).

The three compartments (pods) were made of plastic with non-slip floor mats. On the middle corner of each pod, a small bowl (0.59 L) was fastened over a 1.9 L bowl (stainless steel; Snap’y Fit, Midwest Homes for Pets, Muncie, IN, USA). The smaller bowl was used to offer 100 mL of each solution; the large bowl was used to collect spillage. The maze was set up at the end of the facility, and two gates (described above) were used as the entry way (Figure 2). A standard pig board was used in the main isle to shut the subject in the maze (Figure 2). A camera (GeoVision Model GV-EBL4702-2F; Geovision, Irvine, CA, USA) was placed at a 90° angle over each pod and the entrance area for a total of 4 cameras over the testing arena. A customized NVR recorded each session continuously at 30 frames per second at a video resolution of 1080 p. In real time, one researcher recorded the time of entry and order of entry in each pod. After the session, residual solutions were collected from every bowl and the volume refused was measured and recorded using a 25 or 100 mL graduated cylinder, and then volume consumed was documented as a percentage.

Prior to experiments, pigs were acclimated and conditioned in pairs twice and then individually to properly navigate the maze and 100 mL of +control was available in each pod of the maze (Figure 2). Pigs then were conditioned to predict that there is at least one bowl with desired +control (100 mL). This method causes pigs to explore all 3 options at least once [33]. The location of +control was randomized during this conditioning phase (Figure 2). For the first experiment, each pod randomly contained one of three types of teas. Experiment 1 consisted of 4 identical sessions with the placement of tea type randomized. Four sessions were required because the initial trial was the first time that the pigs were ever exposed to fermented sorghum teas and therefore, novelty responses could be considered. Each session in experiment 1 consisted of a pod containing one of three varieties of sorghum tea on a rotating basis. The personnel caring for the animal were blinded to the treatments, while the personnel conducting the animal portion of the experiment were provided solutions that were labelled with arbitrary numbers and codes.

For experiment 2, three equal parts of the teas from experiment 1 were combined (3Tea; equal portions of white, roasted, sumac) and were tested against the positive and −control (Appendix A). Due to the pigs already being familiar with the fermented tea and controls, only 3 identical sessions were conducted. For each pig and each session, the placement of solutions in the pods were randomized. For both experiments, containers were marked with generic numbers and stickers to improve blindness of the observers, although the tea was obviously a darker color than the control solutions. Control solutions were not dyed with food coloring because food coloring has flavor and pigs use their olfactory senses for preference over vision and food coloring may have additional flavors that pigs can detect.

### 2.4. Video Behavioral Quantification

All videos were collected onto a single drive within the computer system. The videos from the 4 cameras were merged into one file and the Geovision codec was converted into a standard H264 into AVI format. The videos were then manually coded for behaviors using the specialized time stamping (coding) software for interpretive, quantitative analyses (Observer 11.3; Noldus Observer Leesburg, VA, USA). An ethogram with mutually exclusive behaviors was created using previously defined ethograms [34]. Subsequently, a coding system was made within the software. The behaviors included spatial x structural behaviors. Spatial behaviors were defined when the first two legs crossed the threshold (dotted line in Figure 2) and included: entrance, left pod, middle pod, and right pod. Structural behaviors included: head-over-bowl (bowl), exploratory behaviors (i.e., non-nutritive oral behaviors, NNOB; described previously [34]), and all other non-oral and non-exploratory behaviors. Three macros-based keyboards (X-Keys. Engineering, Williamston, MI USA) were programmed to correspond to the behavioral codes in the ethogram. Three trained ethologists coded the same eleven videos within the software. A Cronbach’s reliability test was performed for the 3 observers and 11 samples, and mean alphas were 0.94 ± 0.115 SD and 0.93 ± 0.061 SD for duration and frequency, respectively.

After the agreement test was performed, each trained ethologist was randomly assigned up to 4 pigs to code every session. Ethologists were blinded to the arm treatments for the second conditioning phases and the two treatment phases, and only coded for generic spatial behaviors (e.g., left, middle, right).

### 2.5. Statistical Analysis

Statistical analyses were conducted using SAS version 9.4 (SAS, Cary, NC, USA, SAS Institute Inc.). Data were first transposed in SAS from left, middle, right to correspond to the different choices (Appendix A). A linear mixed model for repeated measures was applied (Restricted-maximum likelihood; ANOVA). Included were the random variable of pigs and the fixed effects of choice (tea type or control added to pod), exposure session (Exp), and the interaction of choice, and exposure and random effect of the pigs were fitted. The covariance structure for both experiments was compound symmetry. The outcome variables included consumption, duration percentage, latency percentage, and rate/number per min. The data were analyzed for normality via Shapiro–Wilk. Consumption data were not normally distributed with and without transformation; therefore, they were evaluated using the non-parametric Kruskal–Wallis test. All other behavior data were square-root-transformed. Least-squares (LS) means for consumed volume, duration, latency, and rate LS means were assessed. Pairwise comparisons were carried out using the two-sided test; the Tukey–Kramer method was used to adjust for multiple comparisons. A treatment difference of *p* ≤ 0.05 was considered significant.

## 3. Results

### 3.1. General Results

The fermented tea types had the following gallic acid concentrations (µg/mL): white, 34.57 ± 0.47 SD; sumac, 140.41 ± 0.41 SD; and roasted sumac, 133.35 ± 0.81 SD. The blended tea (3Tea) had a gallic acid mean of 120.70 ± 11.57 SD µg/mL. During the conditioning phase, one gilt passed away from unknown reasons (necropsy completed by veterinarian and cardiologist) and did not complete experiments 1 and 2.

### 3.2. Experiment 1

No choice by exposure interactions were observed among any variables (*p* > 0.10; Table 1). There was a decrease in consumption volume in the first exposure when compared to later exposures (*p* = 0.005; Table 1). Regardless of tea type, pigs spent the most amount of time performing exploratory behaviors in exposures 1 and 2 (*p* = 0.029; Table 1). Gallic acid intake was the least for the white tea type, and sumac tea type was the greatest, followed by roasted sumac tea type (*p* < 0.001; Table 1).

The rate at which the pigs performed non-oral behaviors was the lowest during the first exposure (*p* = 0.003; Table 1). Pigs performed non-oral behaviors the most (*p* = 0.050; Table 1; Figure 3) when they were exposed to the sumac tea type, and the least when they were exposed to the white tea type.

The pigs checked the bowl at the lowest rate per minute in the first and second exposure. There tended (*p* = 0.060; Table 1) to be a greater latency to express exploratory behaviors during the first exposure.

### 3.3. Experiment 2

Consumption volume was the least for the −control compared to the +control and 3Tea (*p* < 0.001; Table 2).

There was a choice by exposure interaction effect; pigs spent the most time in pods containing the 3Tea blend after the first exposure in experiment 2 (*p* < 0.003; Figure 4).

The total duration spent in the pod was the greatest for 3Tea, and the least for −control (*p* < 0.001; Figure 5). Likewise, pigs spent the greatest duration with their heads over the bowl in the pods with 3Tea, which was greater than +control, and the least was −control (*p* < 0.001; Table 2; Figure 5). The pigs performed more NNOB in the pods with −control (*p* = 0.002; Table 2, Figure 5).

## 4. Discussion

The aim of this study was to determine if the pigs possessed a preference for varying varieties of a fermented sorghum-based tea. This would aid in the development of a satiety product for human consumption. Swine are more biologically relevant subjects for flavor preference in humans than a rodent model [35,36,37]. This is largely due to the decreased phylogenetic distance between humans and swine as opposed to humans and rodents [38,39,40,41]. However, most research involving swine taste preference has been performed in young pigs or lactating sows that are ad libitum fed [42,43,44,45,46].

Some swine researchers cite that sorghum is an inferior grain when compared to corn because growing pigs have a reduced feed intake and average daily gain when fed a sorghum-based diet [47,48]. In swine production, the reputation of sorghum may be based off the fact that fermented sorghum offered to pigs is dried distillers’ grains, a byproduct that could potentially be fed to pigs. Producers are more likely to pick corn-based dried distillers’ grains because of higher levels of metabolizable energy compared to sorghum-based distillers’ grains [49]. Nonetheless, researchers advise that sorghum be properly processed and not be derived from a byproduct in order to increase the bioavailability of nutrients [50,51,52] and the benefits of the antioxidant capacity may improve the immune system [53], which would be more appropriate for limit-fed pigs [54]. Therefore, before the health benefits of fermented sorghum tea can be studied in adult pigs, the authors needed to determine if the sour and bitter flavors of fermented sorghum tea would be rejected. The hypotheses were that pigs would prefer the tea with the least gallic acid, and then prefer the isocaloric control over the blended fermented sorghum tea.

The isocaloric control was created to contain the same amount of sweetness as the fermented sorghum tea. The culture for this tea formula still requires sugar to activate the SCOBY and then they produce fructose and glucose left over as byproducts [55]. The formula also produces byproducts that add tartness. Acetic, gallic, glucuronic, and malic acids after fermentation result in the sourness that can cause rejection among subjects that have not been exposed to these flavors [25,56]. Due to the fermentation process, the fermented tea possesses some carbonation, although humans prefer carbonation [57,58,59]. The final layer commonly reported in the flavor breakdown of SCOBY-fermented teas is that of bitterness and a slight astringency. This can be attributed to a higher than normal level of ethanol that is a naturally occurring byproduct of fermentation, but only 1.0% alcohol by volume of ethanol is legally allowed in commercial fermented teas [56,60]. However, the bitterness and slight astringency are largely attributed to the polyphenol content of the product [56,61].

For the current study, during the acclimation and conditioning phases, the pigs spent a significant amount of time performing non-nutritive oral behaviors, which in the case of test areas, are interpreted as exploratory behaviors. Pigs first had to adjust to the novelty of the three-choice maze test arena, but reduced exploratory behaviors after each session. Then, they had to learn that at least one of three options contained the positive control (isocaloric water sweetened with sucrose), which is a method used for pharmaceutical avoidance or preference research [33].

In the first experiment, three varieties of sorghum tea were evaluated. This first experiment was designed to include four sessions, because the first session would be the first time in the pigs’ lives that they were exposed to a fermented product. As expected, pigs consumed less tea (regardless of type) in the first session compared to the three succeeding sessions. Age influences the amount of time pigs spend in an exploratory state [39,62]. Neophilia increases with age; therefore, in experiment 1, adult pigs displayed less exploratory behaviors and more appetitive behaviors compared to research with young pigs and preference or novelty testing [39,62]. For the first experiment, the only behavioral measure that was influenced by choice was the non-oral behavior duration measure. Non-oral behaviors could be interpreted as refractory, or rather, satiety behaviors [63]. In appetitive behavioral analyses, animals exhibit three states: first, appetitive (i.e., exploratory), then consumption, which is followed by a refractory period [63]. In essence, the consumption of the teas and durations of head-in-bowl were great and similar among all tea types offered, and the non-oral behaviors displayed near the white tea bowl indicate either: (1) a level of satiety, or (2) a greater interest in the sumac-based teas. The authors suspect the latter because gallic acid and other polyphenols are associated with satiety in rodent and human trials [64,65], and the white fermented tea in the current experiment had significantly less gallic acid than the sumac teas. The non-oral behaviors associated with the roasted sumac tea were the same as the non-roasted sumac tea, even though gallic acid was reduced by the roasting process [66,67]. Nonetheless, there are not enough indicators of a preference for a type of tea for the current experiment, but rather, adult pigs display high affinity for all three types of teas based on the fact that they consumed nearly all three teas after the first exposure.

The live observation measures (first choice and volume consumed) did not indicate a clear preference during the trials; therefore, in the second experiment, the authors blended the three teas from the first experiment to compare with the positive control (isocaloric water) and negative control (water). For overall consumption, pigs consumed very little of the negative control, while the isocaloric positive control and blended tea were nearly completely consumed. In addition, the greatest duration of exploratory behaviors were observed near the negative control. For preference research, a negative control is needed to determine if both the positive control and the substrate of interest will be avoided [68]. The refusal volume and exploratory behaviors indicated that the pigs were dissatisfied with the negative control, and avoidance of the blended tea was not observed. Exploratory behaviors include NNOB, which in pigs are also referred to in the literature as oronasal or oral–nasal–facial behaviors because pigs also rub their face and root as a part of NNOB [34,68,69]. These behaviors are commonly studied in sows housed in gestation crates that produce a mentally unstimulating environment. However, the adult pigs in the current experiment were not housed in a mentally stagnant environment and the three-choice maze test added enrichment to their daily activities.

In fact, there were clear indicators that the blended tea was preferred over the isocaloric control, even though they had the same level of calories from sugar. For example, the total amount of time spent in the pod with the blended tea was greater than the positive control, and the total amount of time the head was in the bowl was greater than the positive control, even though the blended tea had a 50-fold increase in gallic acid. As mentioned previously, the acid byproducts from fermentation increases the amount of sour, bitter, and astringent mouthfeel [55,56,70,71]. The results of the current study are in contrast to the assumption that all pigs will reject these added flavors.

Albeit, for the current project, one must note that the flavors which the pigs have shown to avoid are in addition to the sweet flavors from the SCOBY products (i.e., fructose and glucose) [55]. Figuroa et al. [72] compared preference behaviors for umami concentrations (monosodium glutamate—MSG) and compared them to sucrose concentrations in young commercial pigs (1 month old). This work demonstrated that although pigs preferred the highest concentration of umami (MSG), they preferred the medium concentrations of sucrose (4–8%; scaled from 0.5–30% solution). This preferred sucrose concentration is comparable to the current project. The positive control solution was 7.2% sucrose, which matched the Brix-calculated sugar content of the fermented sorghum teas, thus matching Figuroa et al.’s sucrose findings. The authors of this manuscript can deduce that the added flavors from sorghum and fermentation are preferred. Classic pharmaceutical drug avoidance researchers identified bitter flavors without a sweet foundation. Young adult pigs (2–4 months) that were fasted from food and water for 8–16 h had thresholds for the acceptance of bitter compounds when compared to human thresholds [24]. However, more recent researchers identified that water deprivation confounds the preference results [42]. In the current work, our pigs were not water deprived, but rather fed at maintenance. Researchers demonstrated in young pigs that they will prioritize palatability over biological value, which is commonly seen in humans as well [33,41]. For the current research, the polyphenols have biological value, but the added flavors did not appear to phase the adult pigs. In fact, they prefer the fermented sorghum tea over the isocaloric control, which may be an indication that nutrients may be valued in adult pigs.

There is little published research on adult pig preference on flavors [42,44]. Wang et al. [44] examined if the supplementation of the feed of 52 large white lactating sows would impact flavor preference in their piglets. The study was conducted to address the insufficient feed intake of commercial sows during lactation. There were four treatments consisting of an astringent compound such as sodium butyrate, fruit milk flavor, and fruit milk–anise flavor, as well as a control diet. Wang et al. [44] determined that dams consumed for feed when they were treated with the fruit milk–anise. Anise is a seasoning that leaves a bitter after taste [73]; therefore, researchers focused on pre-exposure to flavors via lactation so that piglets have less avoidance behaviors at weaning. The piglets from fruit milk–anise-fed sows increased in feed intake at weaning; therefore, pre-exposure is key to the acceptance of new flavors among young animals. In the current study, the adult pigs had a lack of neophobia for the additional flavors, even though the additional flavors were likely novel. This finding is likely due to the fact that the final product has the foundation of sweetness, but adult pigs that are limit fed are neophilic. Future work will determine if obese pigs have the same affinity to fermented sorghum tea as lean pigs.

## 5. Conclusions

The current study showed that adult pigs have a high affinity to fermented sorghum tea, regardless of tea type. Nonetheless, the main limitation of the current study was in experiment 1; preference (or avoidance) among the three tea types were not determined due to non-significant results. In addition, future work could determine if there is a threshold in sorghum varieties that have very high concentrations of gallic acid, such as black sorghum.

The current work now helps to reduce the assumption that all pigs will reject novel sour and bitter flavors. This research reinforces the legitimacy of the use of swine models for modeling preference in humans and also for improving health in adult pigs. The authors can conclude that future work exploring the health benefits of fermented sorghum tea will not be impeded by a preference of the isocaloric control over the complex high-polyphenol fermented sorghum tea.

## 6. Implications

Sorghum is a highly prolific and sustainable crop that can be grown in many harsh climates, making it a more suitable crop in the face of climate change. Nonetheless, nutritional security and consumer acceptance play a large role in sustainability; if the products produced from a resilient crop are rejected by consumers, then it cannot be considered a sustainable crop. Therefore, sorghum products must be formulated for acceptance, little waste, and improved health. In this preliminary study, fermented sorghum tea was preferred over an equally sweet positive control among adult pigs that were limit fed to maintain a healthy body weight. If future studies indicate that fermented sorghum tea has health benefits, then the evidence of sorghum as a sustainable crop will be strengthened.

## Figures and Tables

**Figure 1 animals-13-03253-f001:**
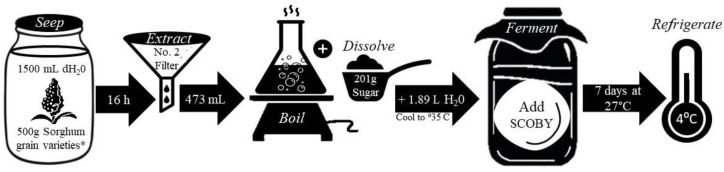
Flow chart outlining the procedure for making fermented sorghum tea using symbiotic culture of bacteria and yeast (SCOBY). * The varieties of sorghum grain included: white, roasted sumac, and sumac.

**Figure 2 animals-13-03253-f002:**
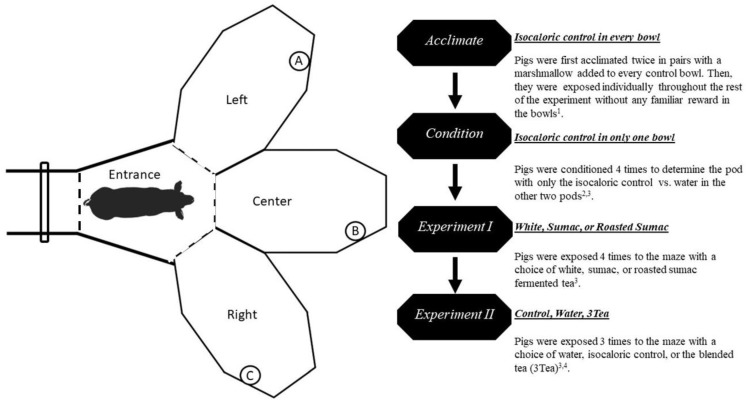
3PodShape Maze diagram: The isle was used as the chute and two standard gates were used to funnel the pig into the center entrance of the three pods (Entrance). The pigs were able to make a choice between a left-angled pod (**A**), a middle pod (**B**), and a right-angled pod (**C**). For daily sessions, the subject entered the entrance and a pig board was used (dashed line) to close the maze. A bowl was placed in each pod and 100 mL of solution was offered. Second Panel: Flow chart representing each step of the preference study. ^1^ The acclimation phase had 100 mL of isocaloric solution (sweet water) in every bowl. After 5 min, pigs that did not explore every pod were led into the area with a marshmallow or a feed pellet was dropped into the bowl. ^2^ The phase did not include leading or extra marshmallows. Three minutes of exploration were allowed. ^3^ The solutions were randomly assigned to a pod (**A**–**C**) so that each pig was exposed to every solution on a rotating basis through each pod ^4^.

**Figure 3 animals-13-03253-f003:**
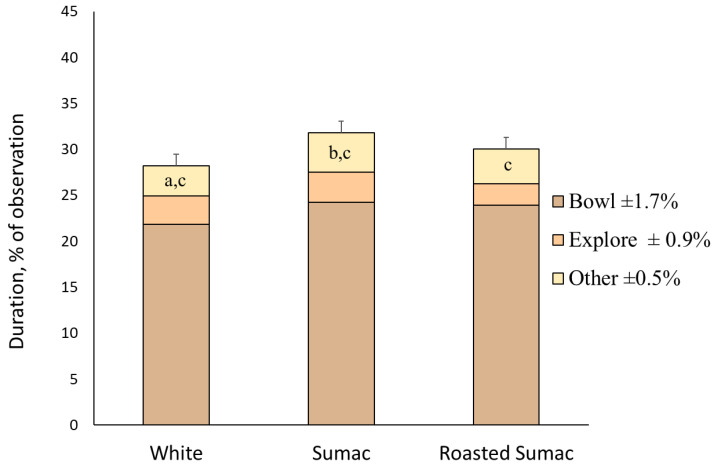
Stack bar graph of pig (n = 11) preference behaviors in a three-choice maze with fermented sorghum tea type. The choices were fermented tea made from white, sumac, or roasted sumac (roasted). The entire bar represents the total duration (% of observation) of time spent in pods with the corresponding solution, with the top error bar representing total duration SEM. The bottom stacks represent the duration of head in bowl, the middle stacks represent the percent duration of exploratory behaviors (i.e., non-nutritive oral behaviors), and the top stacks represent the percent duration of all other behaviors (head still, non-oral, walking or standing). ^a,b,c^ LS means differed (*p* < 0.05).

**Figure 4 animals-13-03253-f004:**
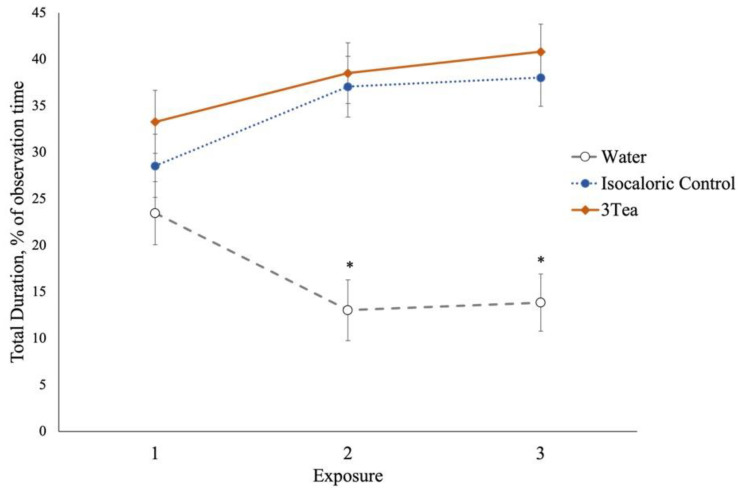
Line graph of sessions for total duration pig (n = 11) preference behaviors in a three-choice maze with fermented sorghum tea type. The choices were: an equal blend of fermented sorghum tea type (3Tea: white, sumac, and roasted sumac), isocaloric solution (−control; sucrose water), or distilled water (+control). The error bars represent SEM. * *p* ≤ 0.05 negative control was less than isocaloric control and 3Tea.

**Figure 5 animals-13-03253-f005:**
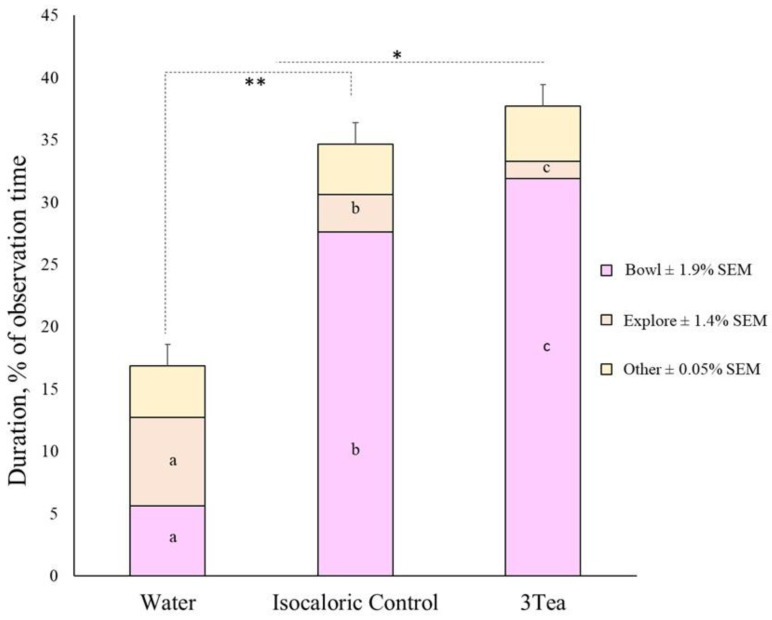
Stack bar graph of pig (n = 11) preference behaviors in a three-choice maze with fermented sorghum tea type. The choices were: an equal blend of fermented sorghum tea type (3Tea: white, sumac, and roasted sumac), isocaloric solution (−control; sucrose water), or distilled water (+control). The entire bar represents the total duration (% of observation) of time spent in pods with the corresponding solution. The bottom stacks represent the duration of head in bowl, the middle stacks represent the percent duration of exploratory behaviors (i.e., non-nutritive oral behaviors), and the top stacks represent the percent duration of all other behaviors (head still, non-oral, walking or standing). * *p* ≤ 0.05 LS means for total duration; LS means between 3Tea and isocaloric; ** *p* ≤ 0.01 for total LS means between −control and the other solutions. ^a,b,c^ *p* ≤ 0.05 LS means within the behavioral category.

**Table 1 animals-13-03253-t001:** Pig (n = 11) preference behaviors in a three-choice maze with fermented sorghum tea type. The choices were fermented tea made from: white, sumac, or roasted sumac (roasted) sorghum. Pigs were exposed (Exp) four times after conditioning.

		Choice					Exposure					*p*-Values			
		White	Sumac	Roasted	SEM ^1^		1	2	3	4	SEM ^1^		Choice	Exp	Choice × Exp
Consumption ^2^, mL		95.2	97.9	95.6	±2.34		87.9^a^	99.9^b^	99.2^b^	97.9^b^	±2.64		0.628	0.005	0.653
Gallic acid, mg		3.3^a^	13.7^b^	12.7^c^	±0.22		9.1^a^	10.3^b^	10.2^b^	10.1^b^	±0.25		<0.001	0.008	0.201
Duration, %															
Total		28.2	31.8	30.0	±1.25		30.1	29.8	30.5	29.7	±1.48		0.208	0.988	0.922
Bowl ^3^		21.9	24.2	23.9	±1.69		23.7	21.0	24.6	24.0	±1.80		0.264	0.213	0.923
Explore ^4,5^		3.1	3.3	2.3	±0.92		3.3^a^	4.7^a^	1.7^b^	1.8^b^	±1.01		0.617	0.029	0.958
Non-oral behaviors ^6^		3.3^a^	4.3^b,c^	3.8^c^	±0.45		3.1^d^	4.0^e^	4.1^e^	3.9^e^	±0.48		0.050	0.097	0.949
Latency, %															
Bowl ^3^		24.4	24.7	26.8	±3.60		30.9	19.3	24.3	26.7	±4.22		0.890	0.327	0.672
Explore ^4^		62.7	67.5	74.1	±6.10		56.6^d^	68.7^e^	75.8^e^	71.1^e^	±6.58		0.192	0.060	0.970
Other ^6^		23.5	23.8	27.5	±7.85		30.8	19.5	23.4	26.1	±8.10		0.696	0.338	0.626
Rate, no./min															
Bowl ^3^		0.68	0.70	0.68	±0.068		0.57^d,e^	0.70^e,f^	0.72^e,f^	0.76^f^	±0.073		0.904	0.069	0.982
Explore ^4^		0.40	0.43	0.32	±0.079		0.47	0.36	0.36	0.35	±0.084		0.302	0.360	0.907
Other ^6^		0.97	1.15	1.08	±0.120		0.81^a^	1.1^b^	1.2^b^	1.2^b^	±0.126		0.173	0.003	0.962

^1^ Largest standard error mean. ^2^ Total volume consumed (100 mL maximum) data were non-parametric when choice was considered. The Kruskal–Wallis test for consumption choice was not significant (χ^2^ = 0.39; *p* = 0.82; df = 2) but was significant for gallic acid (χ^2^ = 97.5; *p* < 0.001; df = 2). ^3^ Head over bowl and head moving indicated drinking. ^4^ Exploratory behaviors (i.e., walking or standing without any sniffing, drinking, chomping, licking, rooting, rubbing using the mouth, snout, or face). ^5^
*p*-Values were derived from square-root-transformed data. ^6^ All non-oral and non-exploratory behaviors. ^a,b,c^ LS means *p* ≤ 0.05. ^d,e,f^ LS means *p* ≥ 0.05 ≤ 0.10.

**Table 2 animals-13-03253-t002:** Pig (n = 11) preference behaviors in a three-choice maze with choices of: distilled water (−control), isocaloric control (+control; water with sugar), or fermented sorghum blended tea (3Tea; equal parts of white, sumac, and roasted sumac). Pigs were exposed (Exp) to the three-choice test three times.

		Choice					Exposure					*p*-Values		
		−Control	+Control	3Tea	SEM ^1^		1	2	3	SEM ^1^		Choice	Exp	Choice × Exp
Consumption ^2^, mL		18.0^a^	96.6^b^	99.0^b^	±2.21		74.1	68.9	70.6	±2.97		<0.001	0.136	0.087
Gallic Acid, mg		0.02^a^	0.20^b^	11.7^c^	±0.03		3.8^a^	3.8^a^	4.3^b^	±0.03		<0.001	<0.001	<0.001
Duration, %														
Total		16.8^a^	34.5^b^	37.7^c^	±1.71		28.4	29.5	31.1	±1.71		<0.001	0.597	0.033
Bowl ^3^		5.6^a^	27.6^b^	31.9^c^	±1.87		20.3	21.8	23.0	±1.92		<0.001	0.365	0.091
Explore ^4,5^		7.1^a^	3.0^b^	1.4^b^	±1.45		4.4	3.5	3.5	±1.50		0.002	0.866	0.261
Non-oral behaviors^6^		4.2	4.1	4.4	±0.45		4.5	4.2	4.0	±0.46		0.583	0.321	0.506
Latency, %														
Bowl ^3^		20.8	16.2	14.8	±3.45		13.93	19.59	18.19	±3.61		0.420	0.481	0.681
Explore ^4^		60.1^d^	71.2^e^	76.0^e^	±7.55		69.0	65.7	72.6	±7.74		0.086	0.608	0.714
Other ^6^		16.2	15.2	13.8	±3.11		12.2	15.6	17.4	±3.26		0.849	0.480	0.958
Rate, no./min														
Bowl ^3^		20.75	16.18	14.77	±3.459		0.42	0.48	0.68	±3.615		0.420	0.481	0.681
Explore ^4^		0.55^d^	0.38^e^	0.35^e^	±0.108		0.39	0.53	0.37	±0.111		0.102	0.196	0.646
Other ^5^		1.41	1.44	1.52	±0.150		1.49	1.46	1.41	±0.152		0.526	0.730	0.413

^1^ Largest standard error mean. ^2^ Total volume consumed (100 mL maximum) data were non-parametric when choice was considered. The Kruskal–Wallis test for choice was significant for consumption volume (χ^2^ = 0.71.6; *p* < 0.001; df = 2) and for gallic acid (χ^2^ = 88.9; *p* < 0.001; df = 2). ^3^ Head over bowl and head moving indicated drinking. ^4^ Exploratory behaviors (i.e., walking or standing without any sniffing, drinking, chomping, licking, rooting, rubbing using the mouth, snout, or face). ^5^
*p*-Values were derived from square-root-transformed data. ^6^ All non-oral and non-exploratory behaviors. ^a,b,c^ LS means ≤ 0.05; ^d,e^ LS means ≥ 0.05 ≤ 0.10.

## Data Availability

Raw data are shared through the USDA-ARS Center for Grain and Animal Health Research publicly archived database.

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
