# Peer review of "Adult Ossabaw Pigs Prefer Fermented Sorghum Tea over Isocaloric Sweetened Water"

_animals, 2023, doi:10.3390/ani13203253_

Round 1
Reviewer 1 Report
The proposal presented by the authors contributes to the development of knowledge in animal biotechnology, the development of foods that improve metabolism in animals and their impact on production are of current interest.
within the discussions it is necessary to expand on these three questions the changes that sorghum undergoes with the proposed methodology as it influences the metabolism of pigs the type of sorghum can influence so that the methodological proposal suffers changes and does not have the results indicated What biochemical changes do sorghum present when using your methodology and how these changes are metabolized or can exert positive or negative effects on the development of pigs?
I suggest the authors modify their conclusion indicating what impact and under what conditions their proposal can be applied and how it impacts at an environmental, social and economic level
Author Response
Reviewer # 1.
R1: The proposal presented by the authors contributes to the development of knowledge in animal biotechnology, the development of foods that improve metabolism in animals and their impact on production are of current interest.
AU: Thank-you, but this first study just aimed to determine if adult pigs would reject a probiotic, fermented sorghum tea.
R1. within the discussions it is necessary to expand on these three questions the changes that sorghum undergoes with the proposed methodology as it influences the metabolism of pigs the type of sorghum can influence so that the methodological proposal suffers changes and does not have the results indicated What biochemical changes do sorghum present when using your methodology and how these changes are metabolized or can exert positive or negative effects on the development of pigs?
AU: We apologize, but this is not a metabolic study, so this information would be outside of the scope of the paper. The tea end product is mainly the polyphenols from sorghum and probiotics. The sponsor does plan to conduct other studies examining the metabolic effects of sorghum grain in pigs.
R1: I suggest the authors modify their conclusion indicating what impact and under what conditions their proposal can be applied and how it impacts at an environmental, social and economic level
AU: The implications do include some information on the overall impact, but this is a fairly novel product therefore there are very limited data to quantify an overall impact.
Reviewer 2 Report
General comment
I appreciate the opportunity to review the present article. I consider it a particularly interesting study regarding nutrition and animal welfare. The article is innovative, and its findings will be valuable for future research. However, prior to its publication it would be necessary to make corrections to improve the understanding of your article and clarity of the information.
Particular comments
Line 2. The title looks interesting. Nonetheless, considering the objective of the study, the authors could modify the title as: “Evaluation of the preference towards .... in adult Ossabaw pigs".
Response:
Lines 17 - 21. This introductory text seems to fit better in the abstract. Consider moving these lines to line 31.
Response:
Line 23. Could the authors indicate the sample size, sex, and average body weight.
Response:
Line 32. Include the sex of the animals.
Response:
Line 33. Could you please include what variables you collected, e.g., the time spent consuming any of the variables or the number of times you approached any of the treatments.
Response:
Line 36- 37. I suggest rephrasing this sentence by mentioning the findings of the present study, and describing whether there were differences between the variables assessed.
Response:
Line 47. Please, revise the Journal’s Instruction for Authors to amend in-text citation style (e.g., instead of “[1,2,3]” it should be “[1 - 3]”. Amend throughout the text.
Response:
Lines 71 - 75. This statement is essential to understand the topic of the present study. Please, consider rephrasing these lines as "In humans, despite the clear benefits on the use of fermented sorghum tea, it is questioned if the same effect could be found in pigs because it is suggested that pigs can show aversion to acid or bitter flavors that may limit its effectiveness. Thus, the relevance of carrying out a palatability study in pigs, for this reason the objective of this article was.... . "
Response:
Line 87. Could the authors indicate how they determined the sample size as significant.
Response:
Line 89. I would recommend that the authors include the inclusion and exclusion criteria in their study.
Response:
Line 117. very good figures that serve as an explanation for the methodology. Thank you for including them.
Response.
Line 147. Correct citation format.
Response.
Line 149. As a recommendation to the authors, I suggest including in a different section called “Experimental design” the explanation on how both experiments were developed. I think this will help to give more order and better clarity to the study.
Response:
Line 150. Could the authors indicate whether this study was prospective or retrospective, blinded or double-blinded.
Response:
Line 164.Could authors please indicate the time of video capture, video resolution, camera angle, and type of camera used?
Response:
Line 171. Revise in text-citation format.
Response:
Line 174. Up to this point, the authors mention that there were observers, when previously they mentioned that the behavioral analysis used software. I invite the authors to clarify if the observers were experienced and to clarify the ethogram used to define the behaviors to be evaluated.
Response:
Line 196 Please, restructure the beginning of the sentence.
Response:
Line 261. Correct dot position.
Response:
Line 300. I agree with this statement and observation. I suggest exploring whether previous bad experiences with degradable flavors might influence food choice of young people.
Response:
Line 370. Include limitations of the present study and future directions or recommendations based on the findings of the study.
Response:
Author Response
eviewer # 2.
R2. I appreciate the opportunity to review the present article. I consider it a particularly interesting study regarding nutrition and animal welfare. The article is innovative, and its findings will be valuable for future research. However, prior to its publication it would be necessary to make corrections to improve the understanding of your article and clarity of the information.
AU: The authors greatly appreciate your thoughtful review. Below is a response (AU) for each item of concern.
Particular comments
Line 2. The title looks interesting. Nonetheless, considering the objective of the study, the authors could modify the title as: “Evaluation of the preference towards .... in adult Ossabaw pigs".
AU: Thank-you for the suggestion. After some consideration, the authors collectively prefer to present an active title rather than the traditional “Evaluation of…” titles. For our research and government funded missions, we are all required to communicate effectively to open-source journals for the public and scientific community. Therefore, our style for the title allows the reader to easily evaluate the usefulness of the manuscript.
Lines 17 - 21. This introductory text seems to fit better in the abstract. Consider moving these lines to line 31.
AU: The authors also prefer an abstract with an introduction sentence. However, the style for Animals requirements are:
- “Abstract: The abstract should be a total of about 200 words maximum. The abstract should be a single paragraph and should follow the style of structured abstracts, but without headings: 1) Background: Place the question addressed in a broad context and highlight the purpose of the study; 2) Methods: Describe briefly the main methods or treatments applied. Include any relevant preregistration numbers, and species and strains of any animals used; 3) Results: Summarize the article's main findings; and 4) Conclusion: Indicate the main conclusions or interpretations. The abstract should be an objective representation of the article: it must not contain results which are not presented and substantiated in the main text and should not exaggerate the main conclusions.
After reviewing previous examples of abstracts published within Animals, we heavily revised the abstract to adhere to the 200 wordcount and restrictions placed in the above bullet point.
Line 23. Could the authors indicate the sample size, sex, and average body weight.
AU: Journals such as JAS do have this standard requirement, but the given the limitations cited above (plus reviewing Animals’ publish manuscripts), the result was an abstract that matched the journal guidelines.
Additionally, the information is specified in section 2.1 of the materials and methods, line 91 and 92.
Line 32. Include the sex of the animals.
AU: Refer to the above AU.
Line 33. Could you please include what variables you collected, e.g., the time spent consuming any of the variables or the number of times you approached any of the treatments.
AU: Refer to the above AU.
Line 36- 37. I suggest rephrasing this sentence by mentioning the findings of the present study, and describing whether there were differences between the variables assessed.
AU: Again, the abstract is limited to 200 words, and the authors need to include the important conditioning component in the abstract.
Line 47. Please, revise the Journal’s Instruction for Authors to amend in-text citation style (e.g., instead of “[1,2,3]” it should be “[1 - 3]”. Amend throughout the text.
AU: Thank-you, please find the highlighted (in yellow) changes throughout.
.
Lines 71 - 75. This statement is essential to understand the topic of the present study. Please, consider rephrasing these lines as "In humans, despite the clear benefits on the use of fermented sorghum tea, it is questioned if the same effect could be found in pigs because it is suggested that pigs can show aversion to acid or bitter flavors that may limit its effectiveness. Thus, the relevance of carrying out a palatability study in pigs, for this reason the objective of this article was.... . "
AU: The authors’ ultimate objectives are to determine health benefits among humans and pigs, therefore, we rephrased these lines and highlighted them within the manuscript to:
Although the final product will be primarily marketed for humans, the current authors anticipate that the byproducts of producing fermented sorghum may also improve health among of commercially produced pigs, particularly among adult pigs that are fed restricted calories to avoid obesity.
Line 87. Could the authors indicate how they determined the sample size as significant.
AU: Sample size calculations were made from a sample size calculator developed by N. E. Elam, New Mexico State Univ., Clayton Livestock Research Center – 2005, which is available on Dr. Mike Galyean’s webpage, an emeritus provost and professor of animal nutrition. This calculator is based off of the formula provided by Kempthorne, O. 1973; The Design and Analysis of Experiments. Krieger, Malabar, FL.
We used the behavioral results that were closest to our target measures, such as: preference (volume intake) for sweetened feed over unsweetened feed; the preference or working memory ratio; duration to complete a memory task; number of visits to each choice in a test, and; the duration of time exploring during a test.
The calculator was set to an alpha of 0.05 and the power (1-beta) to 95. Each measure’s standard error was entered and the desired magnitude of difference between treatments for each measure was entered. The average sample estimation was 5 pigs per treatment. Taking into account that these tests and pig-types are not an exact match four our pilot study, we request an additional pig per treatment. Therefore, this pilot project requests a total of 12 pigs to be tested.
Because this is unique project, the data and variation will be crucial for determining animal numbers on future projects.
Line 89. I would recommend that the authors include the inclusion and exclusion criteria in their study.
AU: Referring to Line 89, it is not clear what is the target concern. Line 89 refers to the animals used on the project. Corvus Scientists (and authors on this manuscript) Drs. Sturek and Alloosh evaluated their herd and selected the appropriate animals for this project. These adult pigs were already housed in a barn together (group housed), but pigs were moved to their randomized pen in a separate barn that was used just for this project.
Line 117. very good figures that serve as an explanation for the methodology. Thank you for including them.
AU: Thank-you! We strive to provide visualization for replication of these studies among other laboratories.
Line 147. Correct citation format.
AU: Changed and highlighted
Line 149. As a recommendation to the authors, I suggest including in a different section called “Experimental design” the explanation on how both experiments were developed. I think this will help to give more order and better clarity to the study.
AU: The intentions of the manuscript were to provide information about the preference fermented sorghum tea in adult pigs rather than provide a technical summary. Nonetheless, this manuscript is part of a Master’s program (first author) which will be completed this year and the technical aspects will be included. Theses are published after committee approval in Krex.
Line 150. Could the authors indicate whether this study was prospective or retrospective, blinded or double-blinded.
AU: The personnel caring for the animal were blinded to the treatments, while the personnel conducting the animal portion of the experiment were provided solutions that were labelled with arbitrary numbers and codes.
Line 164.Could authors please indicate the time of video capture, video resolution, camera angle, and type of camera used?
AU: The camera and videography information were provided in the 2.3 section. These are now highlighted in blue for your references. We did forget the video resolution (thank-you for the reminder), therefore this is provided in this same section, line 146.
Line 171. Revise in text-citation format.
AU: Revised and highlighted.
Line 174. Up to this point, the authors mention that there were observers, when previously they mentioned that the behavioral analysis used software. I invite the authors to clarify if the observers were experienced and to clarify the ethogram used to define the behaviors to be evaluated.
AU: To clarify, this type of software is appropriate for the human observer to review the video and code each behavior (The Observer by Noldus) rather than the use of tracking software (e.g. Ethovision). The corresponding author has extensive experience with Noldus software (22+ years) and has trained the observers to be competent applied ethologists. The ethogram description was reworded for clarity and highlighted in yellow. The reliability tests are highlighted in blue for your reference.
Line 196 Please, restructure the beginning of the sentence.
AU: Change completed and highlighted in yellow.
Line 261. Correct dot position.
AU: This has been amended.
Line 300. I agree with this statement and observation. I suggest exploring whether previous bad experiences with degradable flavors might influence food choice of young people.
AU: The purpose of this statement was to remind the audience that young pigs are neophobic, but the goal of the research is to define tea preference in adult pigs.
Line 370. Include limitations of the present study and future directions or recommendations based on the findings of the study.
AU: The limitations were in experiment 1; this statement is added and highlighted in this section.
Reviewer 3 Report
This is an interesting study, and the manuscript is comprehensible. The study includes two experiments: the first one aimed to evaluate the preference of pigs for a type of fermented sorghum tea; as no preference was found, the follow-up experiment used a mix of all teas to evaluate the relative attraction or repulsion by comparing it to a positive control (water and sugar) and a negative control (water). The results showed that pigs had no aversion for the teas, and consumed most of it during the exposure sessions, to the same level of the positive control, and more than the negative control. Pigs spent more time with the head in the bowl containing the mixture of tea, suggesting that they were more interested in it than in the positive control.
I agree with the authors that this study has value for different fields of research, at the very least for future studies that would want to investigate mildly bitter solutions. However, one must keep in mind that pigs were feed restricted, and that should be better explained in the material and methods, and the implication of the feed restriction on the choice behaviour of the pigs should be discussed! The limitations of the study are very clear to me: pigs seemed to have been free to investigate and drink everything they wanted, and therefore they did. Preference tests usually involve training the animals to locate different resources at FIXED locations, and then letting them choose ONLY ONE of them, the preferred one. Alternatively, animals can be asked to work for a resource, and different amount of work reflect the preference of the pigs (see research by Marian Dawkins: Dawkins, M. S. (1990). From an animal's point of view: motivation, fitness, and animal welfare. Behavioral and brain sciences, 13(1), 1-9.). It does not mean your research is bad, just that it assessed the aversion (or non-aversion) of feed restricted pigs for sorghum tea, but allegations about preference should be avoided (because your design did not allow to assess it in a standard way, and your results do not support them).
Before considering the manuscript for publication, some aspects must be improved:
- The study claims to have been conducted to inform and improve the human health. However, I am unsure that this claim is supported, since the preference of pigs for one type of tea would necessarily translate to human, and a human study on taste preference could have easily been done (i.e. there is no ethical issue).
- Some crucial information about the way the exposure sessions were conducted are missing in the material and methods section, as well as the ethogram detailing the behaviours observed
- The description of the statistical analysis is very poor and contains a lot of mistakes, this should absolutely be revised.
- Some figures do not add to the comprehension of the results, as they duplicate the information provided in the tables.
- Some of the information in the tables need to be better explained.
- Strong allegations about the preference of the pigs should be removed from the conclusion. At best, it is ok to they that the pigs “seemed” to prefer 3Tea over positive control.
See hereunder specific comments and suggestions:
Introduction
L66: do you mean "but as with Kombucha" ?
Material and Methods
L86: Is that the reference of the approval from an ethical committee? Please make it clear.
L87: Why did you only use 11 out of 12 pigs? I saw this information afterwards, but should be in this section
L98-99: Could you state how "dramatic" the feed restriction was? e.g. you could express it as a % of what they would consume if their were fed ad libitum. This is to help the reader understanding the experimental design.
Figure 2
L125: “dotted line” is absent on figure
L126: be consistent : “First Panel:” vs. “Second Panel-“
L130: Superscript 4 is absent in caption but present on picture
L138: The degree symbol is a specific one, not a superscript zero. It can be found in the “Insert Symbol” option in Microsoft Word
L141-143: Just a comment (for future studies): Weighing the bowls could have been better (or an additional measure) to determine the quantity consumed without risk of bias (I believe some liquid remained in the bowls after pouring into the cylinder). Also, how precised was the cylinder used (i.e. in lab it is not advised to use such tools for precise quantifications).
L146: “Pigs were then…” or “Then, pigs were…”
L147: did you check that the pigs indeed visited all three options in the maze in your study ? Over how many sessions (and over how many days) was that stage of habituation performed?
L147: missing "." after the reference – and reference should be a number
L149: Was there only one session per day, or several in one day? When did the session occurred (especially you need to state when it happened relative to the feeding time of the pigs, as it is very important to understand their motivation to eat)? How long were the sessions ? Was there a limited amount of time ? When did you stop the session ? Were the pigs allowed to explore each pod ? Those information are crucial to allow replication of the study.
L153: was that rotation randomised, or the same for each pig ?
L157: was ("the placement")
L170: I guess you mean "front legs" ? Or did you also recorded "entry" when the pig entered backwards?
L171: I guess this is a formatting issue on the reference: it should be “Hulbert et al. (2019)”, or a number
L177: Please be more explicit in how the ethologists were blinded: do you mean that they were unable to see the colour of the solutions (i.e. not visible from the camera angle?) ?
L183: I think you mean GLM ? Because REML correspond to the fitting method for linear models (Restricted maximum likelihood)
L187: What is latency percentage ? The latency to approach one pod, converted in percentage of time of the session ?
L188: Shapiro-Wilk
L188-189: why did you use non-parametric tests? Did the data not meet the normality criteria? You need to explicitly state the reason for choosing those tests.
L190: What do you mean assessed ? LSmeans are produced by GLM and used in pairwise comparisons anyways. I really do not understand your statement (does not add valuable information), also because the next sentence about pairwise comparisons starts with “meanwhile”.
L191: two sided t-test
L191: multiple comparisons, not multiplicity
L192: why did you transformed the p-value ?! Do you (instead) mean you log-transformed the raw data...?
Results
L196: delete “This”
L198-199: cf comment above in M&M - This should be stated in the M&M part. Then what happened to the pen-mate of that gilt ? Did she stay alone for the whole experiment ? I believe this is not ethical to keep a social animal in social isolation over few days.
L202: what do "Choice" and "Exp" refer to? I see it in the Table later, but use the words in full in the text, otherwise that confuses the reader.
L203: It would read better if you say “there was an increase in consumption from first to later exposures” (if you want to imply time-related change), or that “the quantity consumed was lower in the first than in the later exposures”.
L205: Consequently to what ?! The sentence before relates to the behaviour of the pigs...
Table 1: I do not understand this table.... If a pig consumed on average 95.2-97.9 ml of (each) Tea per exposure, how comes that the mean comsumption per exposure in 87.9-99.9 ? You need to explain better the content of the table (i.e. meaning of the variables), it is very confusing at the moment
What is the point of the measure of Gallic acid for the "choice" columns? Gallic acid consumption is related to the tea consumption, and OF COURSE it is higher in more concentrated solutions when their consumption is similar... It makes sense for the "Exposure" column, as I believe this reflects the consumption of all teas within one session (hence the gallic acid content is not linear to the comsumption of tea)
You call “Non-oral behaviours” "other" later in the Table, and in the figure, be more consistent!
I think that information on which arm was entered first (independently of the content of the bowl) would be a valuable information to understand if there was a side bias.
Superscript 5 is missing for the other "explore" (latency, rate), unless it does not apply ? This is not the traditional understanding/definition of "explore" (which usually encompasses sniffing and licking) and I believe you should make your ethogram (definition of each behaviour) clearer, for instance in a dedicated table.
Figure 3: I do not see the point of this figure, as it is the exact same data than in the table 1. I would expect a graphical representation of the consumption of each type of tea per exposure (similar to Figure 4), even if you did not find a significant difference. That would be more informative to the reader, and share more valuable information about the behaviour of the pigs.
L226: That is incorrect. Your table shows that the 3Tea consumption is higher than the -control, but it is the same as +control
Table 2: Please see comments on Table 1 and amend accordingly
Figure 5: Again, I do not see the point of this figure, as all information are already presented in the Table 2
Discussion: To me that section only starts at L295, anything before can be deleted or relocated to the introduction
L258-277: That a complete repeat of your introduction... Discussion should start with the aim of the study, and then your results, not a recall of the introduction !!
L278-L294: This is Material and Methods re-explained, not appropriate for the discussion section which should discuss the results obtained by comparing to other studies, making interpretations of their meaning and their implications, or discussing the flaws of the study…
L299: affective state refers to emotions, exploratory refers to an activity... this sentence is a non-sense
How does this relate to your results ?! the pigs were the same age...
L303: You need to be more precise in which behaviours you observed and categorized as "non-oral behaviours" because not all reflect satiety...
L308: “We” instead of “the authors”? so it is clearer that you are referring to your study and not the cited ones.
L324-325: I think this should be in the introduction (as well, you can re-state it here) in order to understand what "exploration" meant in your experiment.
L332: was greater “than”
L338-339: I do not understand this sentence.
L348: I do not understand this sentence
L367: that is repeating what you just said... should be deleted ("In the current study, the adult pigs had a lack of 366 neophobia for the additional flavors" is enough)
L370: And not feed restricted pigs. I feel that the aspect about the implication of the feed restriction (i.e. higher motivation for food, potential hunger…) are not discussed at all in the discussion, and because there are no information on that in the material and methods (i.e. what 750g of feed represented compared to ad libitum/voluntary food intake), the reader cannot make any interpretation about it. This is a VERY important aspect of your study and it has to be discussed here.
Conclusions
L372: You need to state again that those pigs were feed restricted
L374: You cannot conclude that the high concentrations of gallic acid “improved preference” , as the volume consumed were not different between the three teas, or between the 3Tea and the positive control.
L377-379: I do not see how your study results support that...
Implications
Ok I understand all of that. However, I am not sure your study is justified from an ethical point of view. If the ultimate aim is human consumption, why perform a trial with pigs instead of humans directly ? It seems unethical to use animals as a proxy, when the same study using humans would have carried greated meaning/implications without negative implications for the humans (i.e. they are not forced to drink, and there are no expected negative side effects of this consumption).
I made comments regarding the quality of english along with the comments and suggestions to the authors. There are only a couple of mistakes/typos I could detect.
Author Response
Reviewer # 3.
R3: This is an interesting study, and the manuscript is comprehensible. The study includes two experiments: the first one aimed to evaluate the preference of pigs for a type of fermented sorghum tea; as no preference was found, the follow-up experiment used a mix of all teas to evaluate the relative attraction or repulsion by comparing it to a positive control (water and sugar) and a negative control (water). The results showed that pigs had no aversion for the teas, and consumed most of it during the exposure sessions, to the same level of the positive control, and more than the negative control. Pigs spent more time with the head in the bowl containing the mixture of tea, suggesting that they were more interested in it than in the positive control.
I agree with the authors that this study has value for different fields of research, at the very least for future studies that would want to investigate mildly bitter solutions. However, one must keep in mind that pigs were feed restricted, and that should be better explained in the material and methods, and the implication of the feed restriction on the choice behaviour of the pigs should be discussed! The limitations of the study are very clear to me: pigs seemed to have been free to investigate and drink everything they wanted, and therefore they did. Preference tests usually involve training the animals to locate different resources at FIXED locations, and then letting them choose ONLY ONE of them, the preferred one. Alternatively, animals can be asked to work for a resource, and different amount of work reflect the preference of the pigs (see research by Marian Dawkins: Dawkins, M. S. (1990). From an animal's point of view: motivation, fitness, and animal welfare. Behavioral and brain sciences, 13(1), 1-9.). It does not mean your research is bad, just that it assessed the aversion (or non-aversion) of feed restricted pigs for sorghum tea, but allegations about preference should be avoided (because your design did not allow to assess it in a standard way, and your results do not support them).
To start, there is a major miscommunication that may have caused the entire review to be confounded; Adult pigs, especially pigs that are used as dams and boars MUST be limit fed. In the U.S. it is ethical to follow the National Research Council (NRC) nutrition requirements because obesity among domestic animals and humans is a pandemic. Our pigs were fed 100% of their diet before they were tested in the preference maze.
They were not starving, nor were they fasted. In fact, the Ossabaw research line of pigs was developed for the purpose of evaluating obese and healthy, lean animals. These researchers have extensive knowledge about the health, behavior, and animal welfare of these lines of pigs.
We apologize for this miscommunication, but we did assume that experts in pigs would be reviewing our manuscripts, just as reviewers #1 and #2 understood the meaning of “restricted feed” or are not opposed to using animals for translatable research. Also TWO ethical committees from two different, non-profit institutions (the committees are called “Institution Animal Care and Use Committees”) reviewed our protocol and did not have a problem with approving the protocol. If you would like to learn more about IACUC, you can visit https://www.aalas.org/iacuc
Nonetheless, we added to the introduction a better definition of ‘feed restriction in adult pigs’ so that lay audiences will not be alarmed and we included that each session took place after pigs consumed their daily rations. However, members of society that are opposed to the domestication of animals or their use for research or human consumption (i.e. animal right advocates) will never accept our research, which is their prerogative, but they are not our target audience.
Our preference test was not designed after animal rights philosophy, but rather from previous scientific work on preference among young pigs and sows. This work is completed by applied ethologist, brain scientists, medical doctors, and animal welfare scientists. In our references you will find our resources for designing our experiment.
In this first statement, you mention that the preference maze must be fixed, which conflicts with the comments below, where you ask about randomization. There are plenty of maze examples where the test arena is fixed and the preference for a substrate is measured in our references.
The testing arena is fixed, but the treatments are randomly applied to each pod, and the pigs are the biological beings that evaluate the substrates. Adult pigs can and will refuse substrates that they find boring or off putting, and there is strong scientific evidence that they experience taste like humans, but likely at greater sensitivity due to their olfactory and vomeronasal capabillities. Randomization of the treatments is a standard experimental design for testing food preference in humans and animals. Pig preference for an environment cannot be shuffled around for randomization purposes, therefore, randomization is needed for the order of pigs, and large numbers of animals are needed for these types of studies. Furthermore, our work has dual purposes – both animals and humans will benefit from a low-calorie probiotic with polyphenol product. Therefore, I took the one-dimensional translatable outcomes away in the manuscript, and replaced it with our collective interest in this research.
The citation you provide above insinuates that all animals in captivity are fleeing their oppressive environment and therefore motivation and preference can never be measured in a domestic animal or an animal in the wild that is influenced by human activity. The authors strongly disagree and question the lens of which our manuscript was reviewed.
Furthermore, you ended the review with the comment that we should not use pigs in research, and it is more ethical to study humans. This is perplexing because scientists who do research in animals and humans in our region have strict ethical committees (Institutional Board of Research, IBR). These Review Boards have added to the similar ethics in animal use legal, medical, privacy topics, and other restrictions that tackle the complex history of systemic sexism and racism. If you are against animal research entirely, then you should let the editors know so that they can find articles that are a better fit for your moral standpoint.
Before considering the manuscript for publication, some aspects must be improved:
- The study claims to have been conducted to inform and improve the human health. However, I am unsure that this claim is supported, since the preference of pigs for one type of tea would necessarily translate to human, and a human study on taste preference could have easily been done (i.e. there is no ethical issue).
- Some crucial information about the way the exposure sessions were conducted are missing in the material and methods section, as well as the ethogram detailing the behaviours observed
- The description of the statistical analysis is very poor and contains a lot of mistakes, this should absolutely be revised.
- Some figures do not add to the comprehension of the results, as they duplicate the information provided in the tables.
- Some of the information in the tables need to be better explained.
- Strong allegations about the preference of the pigs should be removed from the conclusion. At best, it is ok to they that the pigs “seemed” to prefer 3Tea over positive control.
See hereunder specific comments and suggestions:
Introduction
L66: do you mean "but as with Kombucha" ?
No. Highlighted change.
Material and Methods
L86: Is that the reference of the approval from an ethical committee? Please make it clear.
Yes. This is also entered in the ethical statement in animals, so it is clear in two places.
L87: Why did you only use 11 out of 12 pigs? I saw this information afterwards, but should be in this section.
No. Outcomes belong in the result section.
L98-99: Could you state how "dramatic" the feed restriction was? e.g. you could express it as a % of what they would consume if their were fed ad libitum. This is to help the reader understanding the experimental design.
As mentioned earlier, this is 100% their required diet to maintain a healthy adult size. Pigs fed ad libitum are fed for growth and/or for fattening.
Figure 2
L125: “dotted line” is absent on figure
Dashed line added
L126: be consistent : “First Panel:” vs. “Second Panel-“
First Panel deleted.
L130: Superscript 4 is absent in caption but present on picture
Superscript changed.
L138: The degree symbol is a specific one, not a superscript zero. It can be found in the “Insert Symbol” option in Microsoft Word
Degree changed
L141-143: Just a comment (for future studies): Weighing the bowls could have been better (or an additional measure) to determine the quantity consumed without risk of bias (I believe some liquid remained in the bowls after pouring into the cylinder). Also, how precised was the cylinder used (i.e. in lab it is not advised to use such tools for precise quantifications).
Noted.
L146: “Pigs were then…” or “Then, pigs were…”
The first.
L147: did you check that the pigs indeed visited all three options in the maze in your study ? Over how many sessions (and over how many days) was that stage of habituation performed?
Yes. See figure 2’s description.
L147: missing "." after the reference – and reference should be a number
Completed change.
L149: Was there only one session per day, or several in one day? When did the session occurred (especially you need to state when it happened relative to the feeding time of the pigs, as it is very important to understand their motivation to eat)? How long were the sessions ? Was there a limited amount of time ? When did you stop the session ? Were the pigs allowed to explore each pod ? Those information are crucial to allow replication of the study.
Figure 2 provides a succinct description of the steps (see reviewer # 2 comments). The additional information for time and sessions are highlighted in yellow.
L153: was that rotation randomised, or the same for each pig ?
Each pig had a different, randomized rotation.
L157: was ("the placement")
L170: I guess you mean "front legs" ? Or did you also recorded "entry" when the pig entered backwards?
Pigs did not enter the test arena backwards (they do not moonwalk). The front legs passing over a landmark is a technical criteria so that this work is repeatable. For example, one may decide that a pig is in a spatial area when the ears cross the threshold. We defined in our M&M the ethogram that the legs were observed as this differed. We want our methods to be refined and repeatable for the scientific community, and defining the exact structure of the criteria is important for replication.
L171: I guess this is a formatting issue on the reference: it should be “Hulbert et al. (2019)”, or a number
Completed change.
L177: Please be more explicit in how the ethologists were blinded: do you mean that they were unable to see the colour of the solutions (i.e. not visible from the camera angle?)
Highlighted changes to observer blindness
L183: I think you mean GLM? Because REML correspond to the fitting method for linear models (Restricted maximum likelihood)
Exposure is a repeated variable that was considered in these analyses. Proc GLM is not suited for repeated measures. Highlighted description of REML.
L187: What is latency percentage? The latency to approach one pod, converted in percentage of time of the session?
Yes, latency in seconds divided by the exact time for the entire session in seconds. Duration behaviors are expressed in percentages, therefore, for consistency, latency is expressed in percentage.
L188: Shapiro-Wilk
Changed
L188-189: why did you use non-parametric tests? Did the data not meet the normality criteria? You need to explicitly state the reason for choosing those tests.
Highlighted change in M&M
L190: What do you mean assessed ? LSmeans are produced by GLM and used in pairwise comparisons anyways. I really do not understand your statement (does not add valuable information), also because the next sentence about pairwise comparisons starts with “meanwhile”.
Highlighted change
L191: two sided t-test
L191: multiple comparisons, not multiplicity
L192: why did you transformed the p-value ?! Do you (instead) mean you log-transformed the raw data...?
This was a typo!! We revised the statement about transforming RAW data. We understand that p-values are not transformed.
Results
L196: delete “This”
L198-199: cf comment above in M&M - This should be stated in the M&M part. Then what happened to the pen-mate of that gilt ? Did she stay alone for the whole experiment ? I believe this is not ethical to keep a social animal in social isolation over few days.
This belongs in the results section, because it is not the original experimental design.
The remaining pig was NOT socially isolated for this short experiment. The pig still had access to interact with the neighboring pigs, especially when manipulating the environmental enrichment. Pigs fight if they are suddenly placed in a new pair or cohort, so this allowed her to get to know her neighbors. This is accepted by our ethical committees, as it is not considered ethical to move a pig to a new pair or group without allowing her time to acclimate to her neighbors through the pen bars. At the end of the study (less than a week later), she was returned to the herd and re-acclimated into a group.
L202: what do "Choice" and "Exp" refer to? I see it in the Table later, but use the words in full in the text, otherwise that confuses the reader.
Choice and exposure are defined in section 2.5 (highlighted blue). Highlighted in yellow is the acronyms.
L203: It would read better if you say “there was an increase in consumption from first to later exposures” (if you want to imply time-related change), or that “the quantity consumed was lower in the first than in the later exposures”.
Your wording is a preference, and not correct as “lower” indicates a height, NOT a quantity.
L205: Consequently to what ?! The sentence before relates to the behaviour of the pigs...
deleted
Table 1: I do not understand this table.... If a pig consumed on average 95.2-97.9 ml of (each) Tea per exposure, how comes that the mean comsumption per exposure in 87.9-99.9 ? You need to explain better the content of the table (i.e. meaning of the variables), it is very confusing at the moment
It is a mixed model based table (TRT, Time, and TRT x Time), the treatment x time interaction is expressed in a line chart when it is significant.
What is the point of the measure of Gallic acid for the "choice" columns? Gallic acid consumption is related to the tea consumption, and OF COURSE it is higher in more concentrated solutions when their consumption is similar... It makes sense for the "Exposure" column, as I believe this reflects the consumption of all teas within one session (hence the gallic acid content is not linear to the consumption of tea).
This is part of the model (as stated previously).
You call “Non-oral behaviours” "other" later in the Table, and in the figure, be more consistent!
See changes in M&M highlighted!
I think that information on which arm was entered first (independently of the content of the bowl) would be a valuable information to understand if there was a side bias.
We did evaluate side bias in the conditioning data, especially when all three bowls had the same contents. We have two statisticians and mathematicians on this project. Side bias was not a factor.
Superscript 5 is missing for the other "explore" (latency, rate), unless it does not apply ? This is not the traditional understanding/definition of "explore" (which usually encompasses sniffing and licking) and I believe you should make your ethogram (definition of each behaviour) clearer, for instance in a dedicated table.
See Hulbert et al. 2019 for the definition of non-nutritive oral behaviors. For pigs, this includes “sniffing, licking, chewing, biting, rubbing using the mouth, snout, or face.” It is widely accepted among applied ethologists as an exploratory behavior among pigs who are exposed to a test arena and to novel substrates. The behaviors are defined in the M&M and the publications cited is a technical paper that visually describes NNOB and the implications.
Figure 3: I do not see the point of this figure, as it is the exact same data than in the table 1. I would expect a graphical representation of the consumption of each type of tea per exposure (similar to Figure 4), even if you did not find a significant difference. That would be more informative to the reader, and share more valuable information about the behaviour of the pigs.
So, you want significant data to not be visually charted, but insignificant data to be charted? This is an unusual preference. We provide the visual representation of the significant data so that the end users can decide for their own purposes if they want the table or the chart.
L226: That is incorrect. Your table shows that the 3Tea consumption is higher than the -control, but it is the same as +control
Changed and highlighted
Table 2: Please see comments on Table 1 and amend accordingly
Change and highlighted
Figure 5: Again, I do not see the point of this figure, as all information are already presented in the Table 2
See response above.
Discussion: To me that section only starts at L295, anything before can be deleted or relocated to the introduction
L258-277: That a complete repeat of your introduction... Discussion should start with the aim of the study, and then your results, not a recall of the introduction !!
No, this section provides further information from other fields (e.g. food science, pig production, sustainability, etc). If you prefer not to provide additional information for your readers in the discussion, that is your preference. We prefer a short, concise introduction and a discussion that provides relevant details. Your rule is a preference, not a policy of Animals.
L278-L294: This is Material and Methods re-explained, not appropriate for the discussion section which should discuss the results obtained by comparing to other studies, making interpretations of their meaning and their implications, or discussing the flaws of the study…
It is appropriate to reiterate that in our experimental design, the very first exposure was completely novel, therefore the results were expected. Reviewer #2 asked that we state limitations of the experiment.
L299: affective state refers to emotions, exploratory refers to an activity... this sentence is a non-sense
Deleted “affective state.” BUT…
Quote:“The “price” an animal is prepared to pay to attain or to escape a situation is an index of how the animal “feels” about that situation.”
Citation: Dawkins, M. (1990). From an animal's point of view: Motivation, fitness, and animal welfare. Behavioral and Brain Sciences, 13(1), 1-9. doi:10.1017/S0140525X00077104
If escape behaviors (an activity) can be referred to as a feeling (affective state), then exploration can also be referred to as a feeling. Or…this cited author is also non-sense?
How does this relate to your results ?! the pigs were the same age...
No line #?
L303: You need to be more precise in which behaviours you observed and categorized as "non-oral behaviours" because not all reflect satiety.
Added more passive language (highlighted) as we are posing different perspectives.
L308: “We” instead of “the authors”? so it is clearer that you are referring to your study and not the cited ones.
Referring to the first person pronouns must be your preference or training. In our training, manuscripts are formal, in the third-person.
L324-325: I think this should be in the introduction (as well, you can re-state it here) in order to understand what "exploration" meant in your experiment.
But you do not want the introduction restated? Be consistent. Or is more than one person serving as Reviwer #3?
L332: was greater “than”
Changed
L338-339: I do not understand this sentence.
Deleted.
L348: I do not understand this sentence
Deleted ‘backdrop….
L367: that is repeating what you just said... should be deleted ("In the current study, the adult pigs had a lack of 366 neophobia for the additional flavors" is enough)
I do not understand this sentence.
L370: And not feed restricted pigs. I feel that the aspect about the implication of the feed restriction (i.e. higher motivation for food, potential hunger…) are not discussed at all in the discussion, and because there are no information on that in the material and methods (i.e. what 750g of feed represented compared to ad libitum/voluntary food intake), the reader cannot make any interpretation about it. This is a VERY important aspect of your study and it has to be discussed here.
Refer to first response.
Conclusions
L372: You need to state again that those pigs were feed restricted
Refer to first response.
L374: You cannot conclude that the high concentrations of gallic acid “improved preference” , as the volume consumed were not different between the three teas, or between the 3Tea and the positive control.
Deleted
L377-379: I do not see how your study results support that...
Ok.
Implications
Ok I understand all of that. However, I am not sure your study is justified from an ethical point of view. If the ultimate aim is human consumption, why perform a trial with pigs instead of humans directly ? It seems unethical to use animals as a proxy, when the same study using humans would have carried greated meaning/implications without negative implications for the humans (i.e. they are not forced to drink, and there are no expected negative side effects of this consumption).
First, refer to first response.
Addition: We also clarified in the introduction (highlighted) that we do expect to further our work for the benefit of both humans and pigs. Does your region not have a human ethics committee, such as institutional review board (IRB)? Human studies require far more ethical considerations and require far more people to do preference tests in even food products. A dozen pigs drinking a probiotic tea after they eat their standard diet has less ethical considerations than human research.
Round 2
Reviewer 2 Report
General comments
I appreciate the authors for following the observations made above for an article that I am convinced is innovative and can add information to the field of study. However, I still consider that they should make minor changes to their manuscript before publication.
Particular comments
Abstract: I understand the concern of the authors to comply with the guidelines guided by the journal on the limitation of the number of words, however, I suggest that relevant data from their study be provided to make the results more interesting for the reader, therefore I invite the authors to make the modifications suggested above that I write below:
Lines 17 - 21. This introductory text seems to fit better in the abstract. Consider moving these lines to line 31.
Response:
Line 23. Could the authors indicate the sample size, sex, and average body weight.
Response:
Line 32. Include the sex of the animals.
Response:
Line 33. Could you please include what variables you collected, e.g., the time spent consuming any of the variables or the number of times you approached any of the treatments.
Response:
Line 36- 37. I suggest rephrasing this sentence by mentioning the findings of the present study, and describing whether there were differences between the variables assessed.
Response:
Line 89. I thank you for your comment and I am sorry for not having explained myself well, I suggest the authors to describe at this level what criteria they followed for the selection and inclusion of the animals and the criteria for the exclusion of the animals from the study.
Response:
Line 91.I appreciate the comprehensive explanation given on the description of the sample size, but, I invite the authors to describe in detail how the sample size calculation was performed because it is a fundamental procedure.
Response:

Author Response
General comments
I appreciate the authors for following the observations made above for an article that I am convinced is innovative and can add information to the field of study. However, I still consider that they should make minor changes to their manuscript before publication.
Particular comments
Abstract: I understand the concern of the authors to comply with the guidelines guided by the journal on the limitation of the number of words, however, I suggest that relevant data from their study be provided to make the results more interesting for the reader, therefore I invite the authors to make the modifications suggested above that I write below:
Lines 17 - 21. This introductory text seems to fit better in the abstract. Consider moving these lines to line 31. Line 23. Could the authors indicate the sample size, sex, and average body weight? Line 32. Include the sex of the animals.
Response: All of the above fit into the re-worked abstract, minus the introductory text. We
Response:
Line 33. Could you please include what variables you collected, e.g., the time spent consuming any of the variables or the number of times you approached any of the treatments. Line 36- 37. I suggest rephrasing this sentence by mentioning the findings of the present study, and describing whether there were differences between the variables assessed.
Response: For the abstract, we can only fit within the 200 word limit the variables that are significant. We did add p-values and the percent duration and percent volume with standard error means. After reworking the abstract to include this information, we had one sentence left for our conclusion.
Line 89. I thank you for your comment and I am sorry for not having explained myself well, I suggest the authors to describe at this level what criteria they followed for the selection and inclusion of the animals and the criteria for the exclusion of the animals from the study.
Response: Included in the M&M is the heard information and criteria for choosing the animals for this experiment. The information is just after the ethical certification, as this is a standard request for our IACUC.
Response:
Line 91.I appreciate the comprehensive explanation given on the description of the sample size, but, I invite the authors to describe in detail how the sample size calculation was performed because it is a fundamental procedure.
Response: Included in the M&M is the heard information and criteria for choosing the animals for this experiment. The information is just after the ethical certification, as this is a standard request for our IACUC.
Reviewer 3 Report
I agree for it to be published, and spotted one typo : L 201 : "square" root transformation (missing word)/
Author Response
Square root transformed was added rather than root transformed.